# Lipid Profiles from Dried Blood Spots Reveal Lipidomic Signatures of Newborns Undergoing Mild Therapeutic Hypothermia after Hypoxic-Ischemic Encephalopathy

**DOI:** 10.3390/nu13124301

**Published:** 2021-11-28

**Authors:** Rebekah Nixon, Ting Hin Richard Ip, Benjamin Jenkins, Ping K. Yip, Paul Clarke, Vennila Ponnusamy, Adina T. Michael-Titus, Albert Koulman, Divyen K. Shah

**Affiliations:** 1The Royal London Hospital, Barts Health NHS Trust, London E1 1FR, UK; rebekah.nixon@nhs.net (R.N.); richard.th.ip@gmail.com (T.H.R.I.); 2Centre for Neuroscience, Surgery and Trauma, Blizard Institute, Barts and the London School of Medicine and Dentistry, Queen Mary University of London, London E1 2AT, UK; p.yip@qmul.ac.uk (P.K.Y.); a.t.michael-titus@qmul.ac.uk (A.T.M.-T.); 3NIHR Core Metabolomics and Lipidomics Laboratory, Wellcome Trust-MRC Institute of Metabolic Science, Addenbrooke’s Hospital, University of Cambridge, Cambridge CB2 0QQ, UK; BJJ25@medschl.cam.ac.uk; 4Neonatal Unit, Norfolk and Norwich University Hospitals NHS Foundation Trust, Norwich NR4 7UY, UK; paul.clarke@nnuh.nhs.uk; 5Norwich Medical School, University of East Anglia, Norwich NR4 7UY, UK; 6Ashford and St. Peter’s Hospitals NHS Foundation Trust, Chertsey KT16 0PZ, UK; vennilaponnusamy@nhs.net

**Keywords:** hypoxic-ischemic encephalopathy, therapeutic hypothermia, newborn, brain, dried blood spot, lipids, lipidome, nutrition

## Abstract

Hypoxic-ischemic encephalopathy (HIE) is associated with perinatal brain injury, which may lead to disability or death. As the brain is a lipid-rich organ, various lipid species can be significantly impacted by HIE and these correlate with specific changes to the lipidomic profile in the circulation. Objective: To investigate the peripheral blood lipidomic signature in dried blood spots (DBS) from newborns with HIE. Using univariate analysis, multivariate analysis and sPLS-DA modelling, we show that newborns with moderate–severe HIE (*n* = 46) who underwent therapeutic hypothermia (TH) displayed a robust peripheral blood lipidomic signature comprising 29 lipid species in four lipid classes; namely phosphatidylcholine (PC), lysophosphatidylcholine (LPC), triglyceride (TG) and sphingomyelin (SM) when compared with newborns with mild HIE (*n* = 18). In sPLS-DA modelling, the three most discriminant lipid species were TG 50:3, TG 54:5, and PC 36:5. We report a reduction in plasma TG and SM and an increase in plasma PC and LPC species during the course of TH in newborns with moderate–severe HIE, compared to a single specimen from newborns with mild HIE. These findings may guide the research in nutrition-based intervention strategies after HIE in synergy with TH to enhance neuroprotection.

## 1. Introduction

Hypoxic-ischemic encephalopathy (HIE) remains an important global cause of death and disability in newborns [1]. Mild therapeutic hypothermia (TH) is shown to be effective in reducing death and disability, with a numbers needed to treat value of seven [2], and is now considered a standard of care [3,4].

Lipids play multiple, key roles, both in the structure, function and homeostasis of the brain [5,6]. The brain contains the largest quantities and the widest diversity of lipid classes and species (Appendix A) of all the organs in the body [5]. Their functions include lipid–protein and lipid–lipid interactions, lipid trafficking, signal transduction, membrane organisation, and energy storage [7]. The neurolipidome, a representation of the lipid content of the brain, is distinct in newborns compared to that of adults and evolves through development [8,9]. It undergoes the greatest change during infancy [8], utilising lipids from the circulation. 

Lipidomic studies provided pathophysiologic insights and biomarker signatures for neurologic disease processes such as motor neurone disease, stroke and dementia [10,11,12,13]. In experimental models of neonatal brain injury, changes in individual lipid species were noted in both hypoxic brain tissue and umbilical cord blood [14,15], e.g., O-phosphocholine, measured by H-NMR, is known to become dysregulated in lipotoxic conditions, such as in ischemia. As such, O-phosphocholine levels in umbilical cord blood were shown to be predictive of HIE severity [15]. 

Dried blood spots (DBS) are routinely used in the first week after birth as part of national newborn screening programmes for the detection of multiple diseases, including cystic fibrosis, congenital hypothyroidism and inborn errors of metabolism [16]. DBS were shown to be a suitable medium for the study of blood biomarkers, as they are relatively non-invasive, easy to transport and easy to store for substantial periods of time [17]. 

Since lipids constitute a sizeable proportion of the brain, are rapidly changing during early life and are integral for its structure and function, we hypothesised that moderate–severe HIE in newborns would lead to the cleavage and the release of membrane lipids, which would then be associated with changes in the concentration of lipids in the circulation. In this study, we aimed to: [1] show that it is feasible to use DBS stored at room temperature to study the lipid profiles of newborns with HIE undergoing TH, [2] assess the changes in lipids between newborns with mild HIE and those with moderate–severe HIE treated with TH, and [3] assess the evolution of lipid profiles during the course of TH. 

## 2. Materials and Methods

Between January 2014 and December 2015, newborns of >36 weeks gestation were recruited as part of the Brain Injury Biomarkers in Newborn Study (BIBiNS) from five neonatal units: The Royal London Hospital (Barts Health NHS Trust, London, UK), Homerton University Hospital NHS Foundation Trust, Ashford and St Peter’s NHS Foundation Trust, University Hospital Southampton NHS Foundation Trust, and Norfolk and Norwich University Hospitals NHS Foundation Trust. The study was approved by a UK research ethics committee (REC ref:13/LO/1738). Newborns were recruited with written consent from parents. 

### 2.1. Participants

Samples were studied from two groups of newborns from this cohort: group (1) newborns admitted to the neonatal intensive care unit with mild acidosis and/or mild HIE (mild HIE group) who did not fulfil standard criteria for TH and were managed conservatively, and group (2) newborns with moderate–severe HIE who fulfilled standard criteria and were treated with TH [18,19]. TH was commenced within 6 h after birth using a servo-controlled total body cooling device maintaining the core temperature as measured by a rectal probe at 33.5 °C for 72 h, after which rewarming was carried out over a period of 12 h.

### 2.2. Blood Sampling and Dried Blood Spots

Group 1 newborns with mild HIE who did not receive TH had a single blood sample taken within 48 h after birth. Group 2 newborns with moderate–severe HIE had blood samples taken at three time points throughout the course of TH: (i) after reaching target temperature (S1), (ii) prior to commencing rewarming (S2), and (iii) after completing rewarming (S3). A drop of blood was collected at each sample point on an absorbent filter paper (Whatman 903 Protein Saver Card) to form a DBS, and the paper was stored in a polythene bag with a desiccant packet, at room temperature. 

### 2.3. Lipid Extraction

The method used for lipid extraction and analysis was previously described in detail [20]. Briefly, using an automated method, lipids were extracted from a single 6 mm chad of each DBS using an Anachem automated liquid handler. Vortexing was carried out, adding 250 μL of internal standard (methanol containing six internal standards: 0.6 μM 1,2-di-*o*-octadecyl-sn-glycero-3-phosphocholine, 1.2 μM 1,2-di-*O*-phytanyl-sn-glycero-3-phosphoethanolamine, 0.6 μM C8-ceramide, 0.6 μM *N*-heptadecanoyl-D-erythro-sphigosyl-phosphorylcholine, 6.2 μM undecanoic acid, 0.6 μM trilaurin) and then adding 750 μL of methyl *tert*-butyl ether (MtBE). Once the solvent extraction was completed, the samples were centrifuged, resulting in two layers, with an aqueous layer at the bottom and an organic layer on top. The extracted lipids from the organic layer were placed in glass-coated, 2.4 mL deep, 384-well plates, along with wells for quality controls (100 μL of H_2_O and 15 μL of plasma) and blanks (200 μL of H_2_O), and stored at −20 °C until further processing. 

The extracted lipid profile was determined using a combined direct-infusion, high-resolution mass spectrometry (DIHRMS) assay. The advantages of DIHRMS were the rapid analysis time, the small amount of sample needed, and the high reproducibility [21]. The limitation of DIHRMS is that it can only separate analytes by mass to charge ratio (*m*/*z*) and this limits identification. Additional liquid chromatography mass spectrometry (LC-MS) on DBS samples enables further identification, based on accurate mass and accurate mass MS^2^ spectra and retention time from a database. Upon lipid extraction and analysis, the raw data were converted to .mzXML files and sent for data analysis.

Eighty-four lipid species were analysed. The lipid species levels were all normalised according to internal standard for their lipid class; therefore, all triglyceride (TG) lipid species were normalised to the TG internal standard. The lipid species levels were expressed as the relative intensity and log2 transformed. 

Then, the fold change difference, which is the ratio of the lipid quantity between two groups studied, was calculated. Therefore, the fold change difference aids in the understanding of upregulation or downregulation of lipid species levels relative to different outcome groups and different time points (S1, S2 or S3). 

### 2.4. Statistical Analysis

In comparing the perinatal characteristics of the two groups of newborns, the Mann–Whitney U test was used for continuous data and the χ^2^ Test [22] or the Fishers Exact Test for categorical data, using SPSS V27.0 (IBM Corp, Armonk, NY, USA). The Fishers Exact Test was used when there was a small sample size with a small number of variables [23].

The raw data from the .mzXML files consisted of all of the DBS lipid profiles. Each lipid specie in the lipid profile was relatively expressed as a percentage of all of the lipid species analysed. With each of the 84 lipid species being measured, the intensity of each lipid species was expressed as a percentage of the intensity of the total lipid profiles for all 84 lipid species for each newborn; termed the relative intensity of the lipid species. 

The distributions of the majority of lipid species were skewed, hence the relative intensities were log2 transformed. K-nearest neighbours imputation (k = 10) was used to impute missing data (package “impute”). All analyses were performed in R 3.6.1 [24]. Principal Component Analysis was then used to identify outliers.

First, univariate analysis was performed by either the Wilcoxon Signed Rank/Rank Sum Test or the Mann–Whitney U test to identify significant differences in relative intensity in each lipid specie (i.e., Mol%).

The Wilcoxon Signed Rank test was used to compare two sets of data that came from the same participants in the study, so it can be used to observe any change in the data from one time point to another [25]. The Mann–Whitney U test was used to compare the differences between two independent samples when they were not normally distributed and the sample sizes were small [26]. Wilcoxon Signed Rank test was used when samples were paired; Mann–Whitney U test was used when samples were not paired. Bonferroni multiple testing correction was applied to all significance thresholds. The Bonferroni corrected significance threshold was used to reduce Type I error when making multiple comparisons within the data by taking the significance threshold (e.g., *p* value < 0.05 or FDR) and dividing it by the number of comparisons [27]. Then, if the multiple univariate analysis showed a significant difference (e.g., *p* value < 0.05 or false discovery rate (FDR)), multivariate analysis was performed. 

Using the “mixOmics” package [28], sparse partial least squares discriminant analysis (sPLS-DA), a supervised statistical test, was utilised here to identify individual lipid species that separate the outcome groups (moderate–severe HIE and mild HIE) and provide a predictive value of the lipid species individually or combined. sPLS-DA selects the most predictive or discriminative features in the data that help classify the samples and allow for variable selection [29]. Area Under the Receiver Operating Characteristic (AUROC) curve was used to determine the model performance by using the lipid species to predict outcome groups. When AUROC curve was 0.7, it meant that there was a 70% chance that model was able to distinguish between the outcome groups [30]. The lower the error rate (range 0–1), the more accurate the model. 

As an extension of the univariate analysis, sPLS-DA was used on the cohort with mild cases and moderate–severe cases at S2 and S3, due to the multiple lipid species identified. 

Considering that nutritional supplementation or the method of feeding of the newborn may be a confounding factor, an additional analysis was performed. Mann–Whitney U test with adjusted *p* value threshold of 0.05 was used. Those in the nil-by-mouth (intravenous dextrose) group were compared to breast-fed, formula-fed and mixed-feeds groups, at each time point. As a final step, those who had parenteral nutrition were compared to those who did not.

## 3. Results

### 3.1. Patients and Samples

Lipid extraction was successful from DBS for the samples from 64 (82%) newborns; a total of 78 samples (21—mild HIE group, 57—moderate–severe HIE group) were processed. The perinatal characteristics of the 64 newborns are shown in Table 1. Newborns in the moderate–severe HIE group were more likely to have a lower Apgar score at 10 min age, a worse base deficit in the first hour, meconium aspiration and seizures. Of the moderate–severe HIE group, eight (17%) newborns had cerebral MRI predictive of an unfavourable outcome [31]. The rating of the MR images was described previously in this cohort [32] using a validated system [31].

### 3.2. Significant Lipid Species Changes between the Single Sample from the Mild HIE Group and the Moderate–Severe HIE Group at S1, S2 and S3

The age of the newborn at the time of the sampling in hours (median age in hours (IQR)) was 23 (15,29) for the mild HIE group. For the moderate–severe HIE group, the time of the sampling in hours after birth (median age in hours (IQR)) for S1 (when newborn had reached the target temperature) was 22 (17,27); for S2 (prior to commencing rewarming) was 59 (50,65); and for S3 (after completing rewarming) was 98 (90,108). Prior to lipid extraction, the DBS were stored at room temperature for up to 4 years. 

At S1, concentrations of one lipid specie, TG (54:5), were higher in mild HIE newborns compared to newborns with moderate–severe HIE (2.031 times higher; 95% CI:1.435, 2.768), FDR: 0.018, (Table 2, Figure 1). At S2, seven lipid species, two phosphatidylcholine (PC) species and five TG species, showed significant changes in relative intensity between the two outcome groups (Table 2, Figure 1); at S3, 29 lipid species exhibited a significant difference between the two groups (Table 3).

### 3.3. Significant Lipid Species Changes over the Course of Therapeutic Hypothermia for Newborns with Moderate–Severe HIE

When comparing lipid levels between S1 and S3, eight lipid species demonstrated significant differences, namely lysophosphatidylcholine (LPC) (15:0), LPC-P (18:1), sphingomyelin (SM) (34:1), SM (36:2), SM (42:3), SM (42:1) and PC (40:5) (Table 4, Figure 1 and Figure 2). When comparing lipid levels between S2 and S3, there was a significant increase in levels of LPC-2O (16:0) and LPC-O (18:1) (Table 4, Figure 2). Furthermore, LPC (15:0), PC (36:2) and PC (36:5) all increased significantly between S1 and S3, while PC-2O (32:0) had decreased significantly by S3 (Table 4).

### 3.4. The Predictive Value of the Significant Changes in Lipid Species in Differentiating Mild HIE Cases from Moderate–Severe HIE Cases Using Sparse Partial Least Squares Discriminant Analysis (sPLS-DA)

As multiple lipid species were identified univariately in groups between mild HIE cases and moderate–severe HIE cases at S2 and at S3, an sPLS-DA model was fitted to investigate the association when considering these lipid species together. The loading scores generated by the sPLS-DA model indicated the degree of confidence in the differential expression of each of the lipid species.

The dataset with mild HIE cases and with moderate–severe HIE cases at S2 was compared in the first sPLS-DA model (Figure 3). The final predictive model gives an AUROC curve of 0.96 (*p* < 0.001, Components 1 and 2) for five of the seven lipid species that were found to be univariately significant, namely TG (50:3), TG (52:4), TG (54:5), PC (34:2) and PC (36:5), the only selected features in the first component (Table 5). In particular, TG (54:5), TG (50:3), and PC (36:5) had moderate-to-high loading scores of 0.745, 0.400 and −0.466, respectively (Table 5), which indicated their significant predictive value within this model.

A second model was fitted using the dataset with mild HIE cases and moderate–severe HIE cases at S3 (Figure 4). The final tuned model gave an AUROC curve of 0.98 (*p* < 0.001, Component 1 and 2) for 28 of the 29 lipid species, that were identified to be univariately significant and selected in the first component (Table 6). The remaining lipid species were selected in the second component. The 28 lipid species represented the majority of the moderate-to-high loading within the first component, again, indicating their significance in the predictive performance (Figure 5).

Using this methodology, upon comparison of the mild HIE cases to moderate–severe HIE cases, one significant species was identified at S1. A further six significant species, as well as the species identified at S1, were noted to be significant at S2 (Table 5) and a further 22 species, plus the previous seven species identified at S2, were noted to be significant at S3 (Table 6, Figure 5).

### 3.5. Nutrition

The nutrition groups of the newborns at S1, S2 and S3 are summarised in Appendix A

When comparing the NBM group to those newborns who were breast-fed (exclusively), formula-fed (exclusively) or mixed at any time point (S1, S2, S3), none of the relative intensities of the lipid species were found to be significantly different (adjusted *p* value threshold <0.05). Furthermore, when comparing the NBM group to different combinations of nutrition supplement (e.g., breast-fed and mixed, formula and mixed, and all three) at any time point, none of the lipid species was found to be significantly different. Finally, when comparing those who had parenteral nutrition and those who did not at any time point, again, none of the lipid species was found to be significantly different.

## 4. Discussion

We demonstrate that it is feasible to extract lipid species in sufficient quantity and quality from a single 6 mm diameter DBS stored at room temperature. DBS is cheap, technically easy to obtain in newborns, and simpler for transportation and storage. Lipids within DBS was shown to be stable for up to a year when stored at −20 °C [17]. In this study we show the feasibility of lipid extraction and analysis from DBS stored at room temperature for up to 4 years. The method used was already validated in newborns [20] and was also used to validate biomarkers of metabolic efficacy in infant nutrition [33]. 

We demonstrate that in newborns with moderate–severe HIE treated with TH, there is an overall reduction in TG and SM classes and an increase in the PC and LPC classes of lipids in comparison to the mild HIE group, in the peripheral circulation, detected using DBS. We also demonstrated that several lipid species differentiate between newborns with mild HIE and those with moderate–severe HIE undergoing TH. This starts with one significant lipid species at S1, then progresses to seven lipid species (six more) at S2, and then to 29 lipid species (22 more) at S3. The most discriminant lipid species are TG (50:3), TG (54:5), and PC (36:5). To our knowledge, this is the first study investigating the lipidome in newborns with HIE through the course of TH. 

### 4.1. The Impact of Nutrition

Based on previous data we expected that nutrition and the type of feed might have confounded some or all of the changes seen in the significant lipid species identified, as infant feeding could alter the lipid profile in DBS [34,35]. In addition, breast milk and each type of formula milk have their own distinctive lipid profiles [36]. All nutrition groups were compared and nutrition was found to have no significant effect on the changes seen in specific lipid species. However, these samples were taken in the first days after birth, and the impact of nutrition on the circulating metabolism might not yet have affected the circulating lipids. In this cohort we showed that the nutrition given to the newborn did not bring about any of the observed significant lipid species changes, and therefore it can be excluded as a confounding factor. 

### 4.2. The Lipid Classes in the Context of What Is Already Known 

Table 7 summarizes lipid changes found in HIE and in adult stroke to add context to our findings.

#### 4.2.1. Triglyceride

TG is an integral energy store in the body, with the liver being an important storage organ. TG (48:1) and (48:2) specifically are the result of de novo lipogenesis [46]. We observed a decrease in the levels of TG species in the moderate–severe HIE at each time point compared to mild HIE. Secondary energy failure is an important mechanism of injury after hypoxia-ischemia (HI) [47], including in newborn HIE [48,49] and in adult stroke [50]. HIE often co-exists with multisystem HI, including in hepatic impairment, as reflected by abnormal liver function tests which also show differences in concentrations between clinical grades of HIE severity [51]. The lower TG levels may be a result of the multisystem HI or of attenuation by TH. If the decreased levels are just a result of temporary attenuation of the increased metabolic demand in ischemia by TH, then the TG levels would be expected to rise at S3 (after rewarming). However, in this study the reduced TG levels persist after rewarming. 

#### 4.2.2. Phosphatidylcholine

We observed increased levels of PC species in the infants with moderate–severe HIE undergoing TH compared to in infants with mild HIE. In an animal model of adult stroke, PC (16:0/18:1) increased in ischemic brain tissue within 2 h of HI and continued to accumulate over 5 days post stroke [43]. In the cord blood after HI, there were increased levels of PC (34:1), PC (36:4), PC (38:4), and PC (38:5) compared to in controls [38,52]. Additionally, during TH, MR-spectroscopy demonstrates decreased levels of PC in the basal ganglia and the white matter in severe HIE compared to in moderate–severe HIE, which may reflect apoptosis in these regions [53]. In experimental models of HI, increased degradation of the glycerophospholipid species, such as PC and PE, were demonstrated in the affected brain tissue at 18–48 h [54,55]. 

We also noted increased levels of PC species at S3 (after rewarming). In an MR-spectroscopy study of newborns with HIE, an increase in PC was shown in the white matter at day 5–6 post TH [53]. We speculate that the increase in PC species upon rewarming (S3) may be associated with the release of PC from damaged cell membranes in the brain. 

We noted that the levels of one PC species differed from how the other PC species behaved. PC-2O (32:0) levels were decreased in moderate–severe HIE compared to mild HIE. PC-2O (32:0) is specifically used in the production of fatty acids in the liver, so its decrease in moderate–severe HIE may be related to secondary energy failure and may also reflect the effect of cooling. 

Phospholipases hydrolyse key groups from phospholipids. Phospholipases are activated in HI through the intracellular Ca^2+^ influx. Activated phospholipases were shown to cause inflammation in HI by releasing arachidonic acid (AA) and prostaglandins from membrane phospholipids, such as PC. In particular, phospholipase A_2_ (PLA_2_) and phospholipase C are significantly activated in HIE, which leads to the production of reactive oxidative species [56,57]. 

#### 4.2.3. Lysophosphatidylcholine

We observed that LPC levels were increased in the peripheral blood of the moderate–severe HIE compared to the mild HIE. We speculate that this might reflect a compensatory mechanism to attenuate injury from HI. It is possible that an LPC-rich diet to increase peripheral levels might prove to be neuroprotective. 

We speculate that the lower levels of LPC at S2 compared to S3 (rewarming) over the course of TH might be due to the effect of TH, reducing metabolic rate, followed by an increase in LPC after rewarming which may reflect the wearing off of the effect of TH. 

The interpretation of the LPCs analysed from DBS is challenging as lipases may remain active during the drying process [58] and some hydrolysis can even occur during storage. These processes are unlikely to influence the results here as for each newborn the samples were taken within days after birth and the total storage time was relatively long (up to 4 years).

#### 4.2.4. Sphingomyelin

SM species, especially SM (36:1) and SM (36:2) are important constituents of the myelin sheath [5,59]. Some SM species are in the outer layer of the lipid bilayer of cell membranes. After cerebral ischemia, there is compromise of the blood–brain barrier, so neurolipids may be found in the peripheral blood [60].

In our study SM (36:2) was found to be one of the better discriminant lipid species for the severity of HIE. We observed decreased levels of all significant circulating SM species, including SM (36:2), in the moderate–severe HIE group compared to the mild HIE group. To our knowledge this is the first study of SM levels in newborn HIE. Our findings are contrary to evidence from animal stroke models. In animal neonatal HIE and adult stroke models, there were decreased SM levels, specifically SM (18:0), in ischemic brain tissue associated with apoptosis and the demyelination of the ischemic tissue [43,54,55]. In alignment with this, in an animal stroke model an increase in plasma SM (37:1) and SM (38:3) levels was noted [44].

Acid sphingomyelinase (ASM) hydrolyses SM into ceramide (Cer), which is overactivated in ischemia [61]. It was suggested that in ischemia there is a breakdown of SM into Cer, causing an accumulation of Cer, which is a potent inducer of apoptosis [62,63]. Interestingly, despite including Cer in our extraction and analysis, we did not find any significant difference in the levels of the Cer species between mild and moderate–severe HIE at any time point. With the decrease in SM, one would expect a downstream increase in Cer.

### 4.3. Strengths

In this study, a validated and reliable method to extract lipid species from DBS was used [20]. Several statistical approaches, including univariate analysis, multivariate analysis and sPLS-DA models were combined in order to provide confirmation of the lipid species that showed significant changes. In comparison with the mild HIE group, additional significant lipid species were identified through the course of hypothermia and rewarming, but the lipid species identified at the previous sampling time point were still retained at later time points. Our study identified predictive trends in the entire class of lipids during TH. Any potential confounding effect of the nutritional status of the newborn on the lipid profiles was also adjusted for. 

### 4.4. Limitations

The sample timing of the DBS collection was variable and samples were obtained when convenient in the clinical setting to minimise painful venesection. We were able to extract lipids successfully from 82% of samples. Eight (17%) of the forty-six newborns with moderate–severe HIE who underwent TH had substantial brain injury in MRI, predictive of an adverse outcome. As such, the study was not able to compare the neurolipidome between newborns with and without substantial brain injury in MRI. 

A study such as ours can only determine differences and associations and not causation. It is important to consider throughout the interpretation of the data that the changes we showed in moderate–severe HIE cases managed with TH were likely to be a summation of the pathophysiology and of the therapeutic intervention; and discerning the exact contribution of each of these factors remains a significant challenge. 

## 5. Conclusions

Using DBS, this study demonstrates a reduction in TG and SM and an increase in the PC and LPC classes of circulating lipids during the course of TH in newborns with moderate–severe HIE, compared to a single specimen from newborns with mild HIE. Larger studies are required to study the neurolipidome changes in newborns with substantial brain injury. These methods and findings may guide the research in nutrition-based intervention strategies after HIE, in synergy with TH, to enhance neuroprotection.

## Figures and Tables

**Figure 1 nutrients-13-04301-f001:**
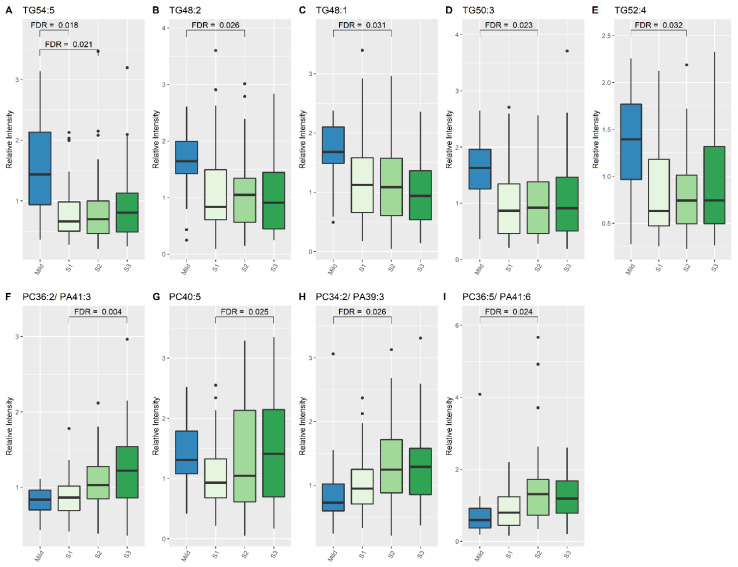
Lipid species from TG and PC classes which show significant differences between Mild HIE Cases and Moderate–severe HIE Cases at different time points (S1, S2 and S3). (**A**), TG54:5; (**B**), TG48:2; (**C**), TG48:1; (**D**), TG50:3; (**E**), TG52:4; (**F**), PC 36:2/PA41:3; (**G**), PC40:5; (**H**), PC 34:2/PA39:3; (**I**), PC 36:5/PA41:6.

**Figure 2 nutrients-13-04301-f002:**
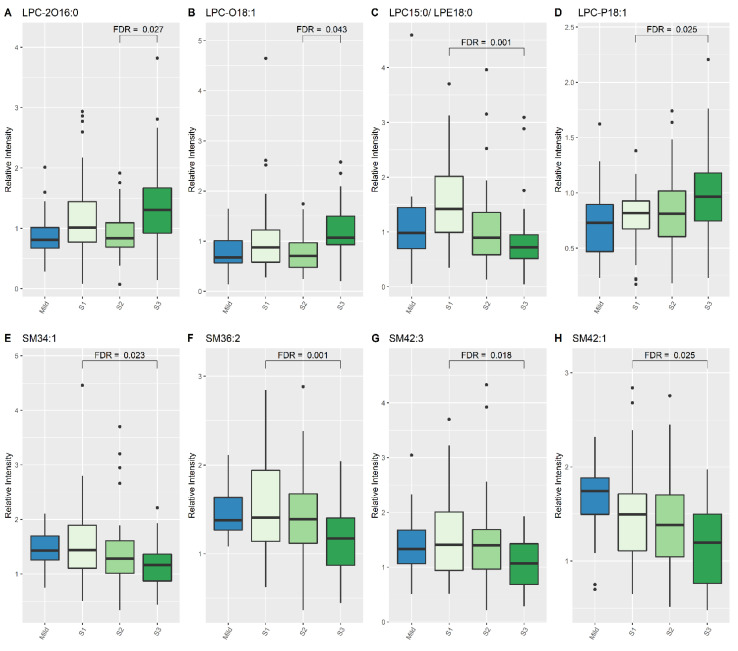
Lipid species from LPC and SM classes which show differences between Mild HIE cases and Moderate–severe HIE cases at different time points (S1, S2 and S3). (**A**), LPC 2016:0; (**B**), LPC 018:1; (**C**), LPC 15:0/LPE18:0; (**D**), LPC P18:1; (**E**), SM34:1; (**F**), SM36:2; (**G**), SM42:3; (**H**), SM42:1.

**Figure 3 nutrients-13-04301-f003:**
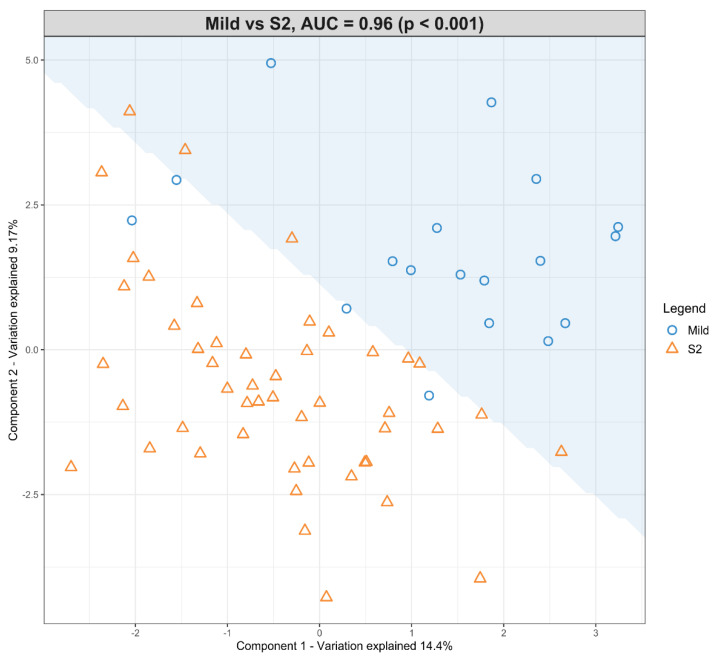
Scatter plot showing the sPLS-DA model and its prediction for Mild cases and Moderate–severe cases at S2. The *x* axis represents Component 1 and the *y* axis represents Component 2.

**Figure 4 nutrients-13-04301-f004:**
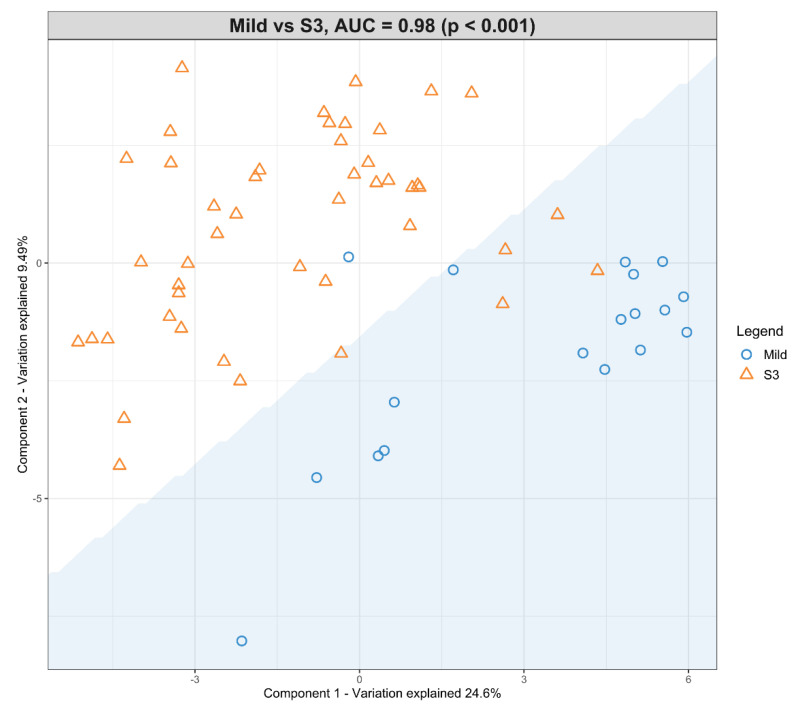
Scatter plot showing the sPLS-DA model and its prediction for Mild cases and Moderate–severe at S3. The *x* axis represents Component 1 and the *y* axis represents Component 2.

**Figure 5 nutrients-13-04301-f005:**
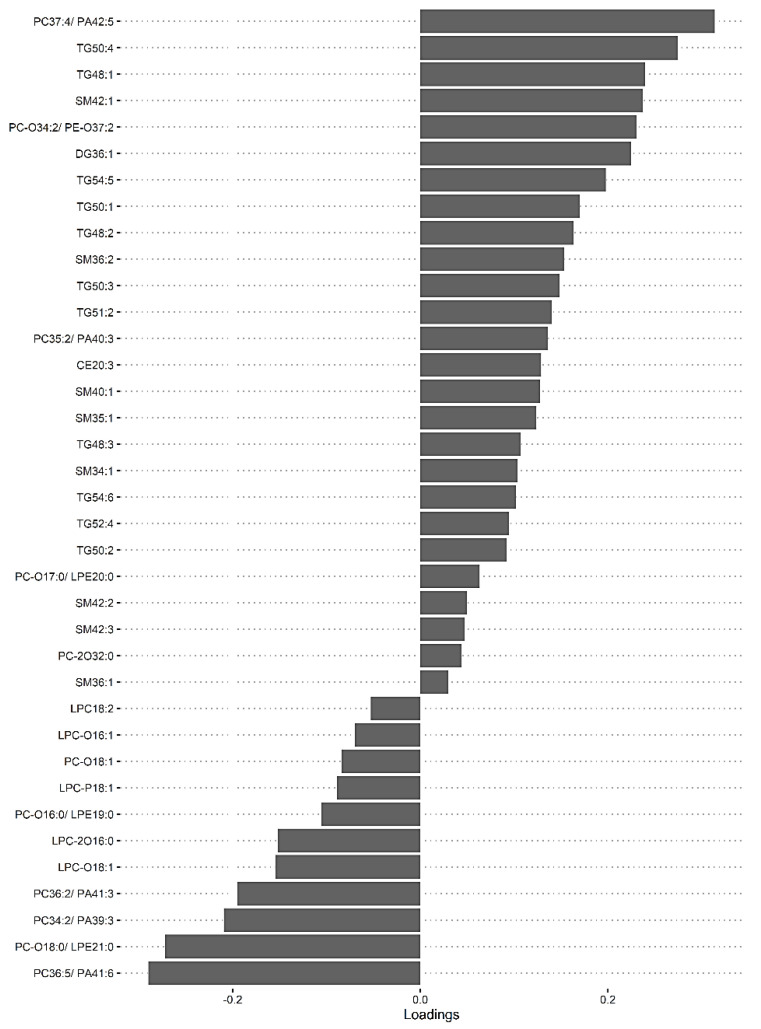
sPLS-DA model of Mild HIE cases vs. Moderate–severe HIE cases at S3, loadings plot of Component 1.

**Table 1 nutrients-13-04301-t001:** Baseline characteristics of the studied population.

Perinatal Characteristics	Mild HIE	Moderate–Severe HIE	*p* Value
	No TH	TH	
*n*	18	46	
Male	5 (28%)	25 (54%)	0.06
Birth weight (g)	3520 (3080, 3900)	3450 (3105, 3860)	0.81
10 min Apgar score	9 (8, 10)	6 (5, 7)	<0.001 *
Chest compressions	2 (11%)	9 (20%)	0.48
Inotropes used	0 (0%)	16 (35%)	0.003 *
Worst pH in first hour	6.95 (6.93, 7.00)	6.92 (6.84, 7.00)	0.24
Worst base deficit in first hour	−13.2 (−14.0, −10.9)	−16.2 (−18.0, −14.9)	<0.001 *
Sentinel event	6 (33%)	11 (24%)	0.54
Meconium aspiration	0 (0%)	9 (20%)	0.05 *
Seizures	0 (0%)	25 (54%)	<0.001 *
Unfavourable MRI	-	8 (17%)	-

* -denotes significance where *p* < 0.05. Values stated as median (interquartile range).

**Table 2 nutrients-13-04301-t002:** Mann–Whitney U test statistics for significant lipid species when comparing Mild vs. Moderate–severe HIE at S1 and S2.

*Moderate–Severe HIE at S1* vs. *Mild*					
Lipid Species	Moderate–Severe HIE Lipid Median	Mild HIE Lipid Median	Change in Relative Intensity as a Multiple (95% Confidence Interval)	*p* Value	Adjusted *p* Value (FDR)
TG (54:5)	0.66	1.432	2.031 (1.4345, 2.768)	<0.001	0.018
***Moderate–Severe HIE at S2* vs. *Mild* **					
**Lipid Species**	**Moderate–Severe HIE Lipid Median**	**Mild HIE Lipid Median**	**Change in Relative Intensity as a Multiple (95% Confidence Interval)**	** *p* ** **Value**	**Adjusted** * **p** * **Value** **(FDR)**
PC (34:2)/PE (37:2)/PA (39:3)	1.247	0.726	0.616 (0.487, 0.829)	0.002	0.026
PC (36:5)/PE (39:5)/PA (41:6)	1.31	0.589	0.494 (0.327, 0.736)	0.001	0.024
TG (48:1)	1.087	1.678	1.542 (1.183, 2.061)	0.002	0.031
TG (48:2)	1.049	1.642	1.605 (1.236, 2.214)	0.001	0.026
TG (50:3)	0.92	1.628	1.708 (1.267, 2.493)	0.001	0.023
TG (52:4)	0.741	1.394	1.691 (1.219, 2.253)	0.003	0.032
TG (54:5)	0.695	1.432	2.018 (1.447, 2.836)	<0.001	0.021

Abbreviations: FDR—false discovery rate.

**Table 3 nutrients-13-04301-t003:** Mann–Whitney U test statistics for the significant lipid species when comparing Moderate–severe HIE at S3 vs. Mild HIE.

Lipid Species	Moderate–Severe HIE Lipid Median	Mild HIE Lipid Median	Change in Relative Intensity as a Multiple (95% Confidence Interval)	*p* Value	Adjusted *p* Value (FDR)
PC (34:2)/PE (37:2)/PA (39:3)	1.291	0.726	0.631 (0.479, 0.825)	0.001	0.012
PC (35:2)/PE (38:2)/PA (40:3)	0.761	1.119	1.509 (1.171, 1.947)	0.004	0.017
PC (36:2)/PE (39:2)/PA (41:3)	1.125	0.838	0.725 (0.5910, 0.872)	0.003	0.015
PC (36:5)/PE (39:5)/PA (41:6)	1.186	0.589	0.495 (0.351, 0.692)	0	0.006
PC-O (16:0)/LPE (19:0)	1.209	0.595	0.552 (0.3784, 0.862)	0.014	0.045
PC-O (18:0)/LPE (21:0)	1.238	0.533	0.505 (0.361, 0.717)	0.001	0.011
PC-2O (32:0)	1.275	1.518	1.182 (1.094, 1.303)	0.001	0.011
PC-O (34:2)/PE-O (37:2)	0.837	1.498	1.770 (1.294, 2.445)	0.002	0.015
PC-O (34:1)/PE-O (37:1)	1.283	1.52	1.175 (1.053, 1.318)	0.008	0.032
PC (37:4)/PE (40:4)/PA (42:5)	0.692	1.525	2.073 (1.430, 2.990)	0	0.006
LPC-O (18:1)	1.094	0.677	0.619 (0.493, 0.855)	0.002	0.015
LPC-P (18:1)	0.976	0.726	0.7195 (0.553, 0.932)	0.017	0.05
SM (34:1)	1.266	1.429	1.235 (1.029, 1.471)	0.017	0.05
SM (35:1)	1.118	1.765	1.403 (1.122, 2.015)	0.004	0.019
SM (36:2)	1.185	1.38	1.219 (1.061, 1.433)	0.01	0.035
SM (40:1)	1.225	1.624	1.361 (1.127, 1.778)	0.003	0.015
SM (42:1)	1.221	1.741	1.405 (1.188, 1.724)	0	0.006
DG (36:1)	0.794	1.652	1.968 (1.368, 2.866)	0.002	0.013
TG (48:1)	0.939	1.678	1.727 (1.306, 2.512)	0	0.006
TG (48:2)	0.909	1.642	1.646 (1.180, 2.667)	0.002	0.015
TG (48:3)	0.81	1.662	1.735 (1.152, 2.476)	0.011	0.038
TG (50:1)	0.841	1.775	1.929 (1.341, 2.821)	0.003	0.015
TG (50:2)	0.847	1.459	1.693 (1.100, 2.365)	0.011	0.038
TG (50:3)	0.91	1.628	1.631 (1.163, 2.450)	0.004	0.017
TG (50:4)	0.92	1.828	1.722 (1.319, 2.294)	0.001	0.009
TG (51:2)	0.74	1.149	1.613 (1.116, 2.110)	0.009	0.034
TG (52:4)	0.744	1.394	1.535 (1.068, 2.224)	0.016	0.049
TG (54:5)	0.804	1.432	1.739 (1.232, 2.462)	0.003	0.015

Abbreviations: FDR—false discovery rate.

**Table 4 nutrients-13-04301-t004:** Wilcoxon Signed Rank test statistics of the Moderate–severe HIE newborns when comparing different time points during therapeutic hypothermia (S1, S2 and S3).

*S1* vs. *S3 for Moderate–Severe HIE*					
Lipid Species	Median Level at Birth	Median Level at Re-Warming	Change in Relative Intensity as a Multiple (95% Confidence Interval)	*p* Value	Adjusted *p* Value (FDR)
PC (36:2)/PE (39:2)/PA (41:3)	0.865	1.222	0.711 (0.599, 0.832)	<0.001	0.004
PC (40:5)	0.932	1.409	0.576 (0.459, 0.832)	0.002	0.025
LPC (15:0)/LPE (18:0)	1.42	0.717	1.935 (1.456, 2.442)	<0.001	0.001
LPC-P (18:1)	0.818	0.964	0.811 (0.713, 0.932)	0.002	0.025
SM (34:1)	1.437	1.163	1.443 (1.174, 1.729)	0.001	0.023
SM (36:2)	1.407	1.171	1.382 (1.22, 1.580)	<0.001	0.001
SM (42:1)	1.496	1.197	1.402 (1.126, 1.667)	0.002	0.025
SM (42:3)	1.405	1.067	1.535 (1.193, 1.956)	0.001	0.018
***S2* vs. *S3 for Moderate–Severe outcome HIE***					
**Lipid Species**	**Median Level at Cooling**	**Median Level at Re-Warming**	**Change in Relative Intensity as a Multiple (95% Confidence Interval)**	***p* Value**	**Adjusted *p* Value** **(FDR)**
LPC-2O (16:0)	0.837	1.308	0.693 (0.554, 0.847)	<0.001	0.027
LPC-O (18:1)	0.704	1.073	0.647 (0.519, 0.826)	0.001	0.043

Abbreviations: FDR—false discovery rate.

**Table 5 nutrients-13-04301-t005:** sPLS-DA loading values for Mild HIE newborns vs. Moderate–severe HIE newborns at S2.

Lipid Species	Component 1	Component 2	Component 3
PC (34:2)/PE (37:2)/PA (39:3)	−0.13725	0	0
PC (36:5)/PE (39:5)/PA (41:6)	−0.46582	−5.03 × 10^4^	0
TG (48:1)	0	0	0
TG (48:2)	0	0	−0.12742
TG (50:3)	0.400286	0	0
TG (52:4)	0.221072	0	0
TG (54:5)	0.745032	0	0

**Table 6 nutrients-13-04301-t006:** sPLS-DA loading values for Mild HIE cases vs. Moderate–severe HIE Cases at S3.

Lipid Species	Component 1	Component 2	Component 3
PC (34:2)/PE (37:2)/PA (39:3)	−0.20961	0.127569	0
PC (35:2)/PE (38:2)/PA (40:3)	0.135575	0	0
PC (36:5)/PE (39:5)/PA (41:6)	−0.29015	0.141899	0
PC (36:2)/PE (39:2)/PA (41:3)	−0.19513	0.168625	0
PC (37:4)/PE (40:4)/PA (42:5)	0.313908	0	0
PC-O (16:0)/LPE (19:0)	−0.10542	−0.11515	0
PC-O (18:0)/LPE (21:0)	−0.27236	0	0
PC-O (34:2)/PE-O (37:2)	0.230711	0	0
PC-O (34:1)/PE-O (37:1)	0	0.069747	0
PC-2O (32:0)	0.043895	0	0
LPC-O (18:1)	−0.15428	0	0
LPC-2O (16:0)	−0.15171	0	0
LPC-P (18:1)	−0.08909	0.067468	0
SM (34:1)	0.103771	0	0.484361
SM (35:1)	0.123691	0.015681	0
SM (36:2)	0.153531	−0.06324	0
SM (40:1)	0.127328	0	0
SM (42:1)	0.237286	0	0
DG (36:1)	0.224861	0	0
TG (48:1)	0.239505	0	0
TG (48:2)	0.16305	0	0
TG (48:3)	0.106817	0.044888	0
TG (50:1)	0.169699	0	0
TG (50:2)	0.091848	0	0
TG (50:3)	0.14831	0	0
TG (50:4)	0.274731	0	0
TG (51:2)	0.14014	0	0
TG (52:4)	0.094162	0.01366	0
TG (54:5)	0.197721	0	0

**Table 7 nutrients-13-04301-t007:** Lipid analysis on animal models of and human studies of adult stroke and HIE Abbreviations: MCAO—middle cerebral artery occlusion.

Reference	Biofluid/Tissue	Species and Population	Discriminant Lipid Species and Observed Changes in Disease Group
HIE			
Liu et al. 2013 [37]	Whole brain extract	Mice—6 no reoxygenation, 6 controls, 6 hypothermia, 6 normothermia, 6 rewarming, 6 long normothermia	**Increase**: choline and PC
Walsh et al. 2012 [38]	Umbilical cord blood	Human newborns—40 asphyxia vs. matched controls Human newborns—31 HIE vs. matched controls	**Increase**: LPC (16:0), PC (34:1), PC (36:4), PC (38:4) and PC (38:5) **Increase**: PC (38:4)
Reinke et al. 2013 [15]	Umbilical cord blood	Human newborns—34 Asphyxia vs. matched control Human newborns—25 HIE vs. matched controls	**Increase**: choline and O-phosphocholine **Increase**: choline
**Adult Stroke**			
Yang et al. 2017 [39]	Plasma	Human adults—29 lacunar infarct patients vs. 15 healthy controls	**Decrease**: PE species, especially PE (35:2) **Increase**: LPC species—LPC (20:4), LPC (20:5), LPC (22:6) and LPC (24:0). TG species, especially TG (56:5)
Koizumi et al. 2010 [40]	Brain tissue sections	Sprague–Dawley Rats—MCAO model of acute stroke vs. normal brain sections	**Decrease**: PC (16:0/18:1) due to overactivation of PLA_2_ **Increase**: LPC (16:0)
Irie et al. 2014 [41]	Whole brain sections	Male Wistar Rats—MCAO model of stroke vs. healthy controls	**Decrease**: CDP-choline and PE species at reperfusion
Wang et al. 2010 [42]	Cerebral cortex	Rats—MCAO model of acute stroke	**Decrease**: PC species and SM species **Increase**: LPC species due to lipid breakdown and neuroinflammation.
Nielsen et al. 2016 [43]	Whole brain coronal slices	Male C57BL/6 male Mice—MCAO model of stroke vs. healthy controls	**Increase during ischemia**: LPC species, especially LPC (16:0) in acute ischemia, PS species and NAPE ^1^ species **Decrease during ischemia**: SM species decreased and disappeared by 24 h **Increase during reperfusion**: spingosine-1-phosphate and DHA
Sheth et al. 2015 [44]	Rodent whole brain slices and plasma from humans	MCAO stroke model in mice vs. TBI model in rats Plasma from 9 human patients with acute ischemic stroke vs. 5 patients with stroke mimics and no stroke	**Increase**: within 24 h of stroke sharp increase in the sphingolipid (SL) score, using SM (36:0) and Cer (42:1) species with BBB disruption. The SL score correlates with the volume of infarct.
Lind et al. 2020 [45]	Plasma	Human adults—3 independent populations of ischemic stroke	**Decrease**: SM (32:1) in ischemic stroke. The lower the level of SM (32:1), the higher the rate of incident ischemic stroke.

^1^*. N*-acyl-phosphatidylethanolamine species.

## Data Availability

The data presented in this study are available on request from the corresponding author.

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
