# Peer review of "Lipid Profiles from Dried Blood Spots Reveal Lipidomic Signatures of Newborns Undergoing Mild Therapeutic Hypothermia after Hypoxic-Ischemic Encephalopathy"

_nutrients, 2021, doi:10.3390/nu13124301_

Round 1

Reviewer 1 Report

This study investigates the feasibility and reports first results of the analysis of lipidomic profiles from dried blood spots in newborns with mild and moderate to severe hypoxic-ischemic encephalopathy treated with hypothermia.

This work is original and may have a considerable impact. 

The paper is well written. The title is quite long, but I don't see any way to shorten it.

The abstract is concise and meaningful. I only doubt whether the last sentence is supported by the results (cf below).

The introduction is short and relevant with the aims of the study clearly stated.

Material and methods are appropriately described. Obviously, the number of patients was not based on a power calculation but on a collecting period of two years. The criteria for patient selection and the modalities of body cooling are adequately described. The method used for lipid extraction and analysis as well as the statistics used are well described as far as a non-specialist can judge.

The results are extensively given. I wonder whether the same facts should be presented in tables and figures. The differences in lipid profiles between the two groups are significant and relevant. However, the authors could not demonstrate a statistically significant difference between the various feeding regimes, probably because the study was underpowered for this goal.

Therefore I doubt whether the conclusion in regard to nutrition based strategies to treat HIE is justified.  The statement “these findings may guide interventional strategies….” are rather based on general assumptions and animal experiments than on results obtained in this study.

Reviewer 2 Report

This research seems to me very interesting but very descriptive in terms of the observed changes in lipids As for the conclusion, I would like to know what changes in diet should be recommended or what direction to take This work has the merit to exist and can be published if there is an agreement of the rewievers
